# The Associations between Perioperative Blood Transfusion and Long-Term Outcomes after Stomach Cancer Surgery

**DOI:** 10.3390/cancers13215438

**Published:** 2021-10-29

**Authors:** Fu-Kai Hsu, Wen-Kuei Chang, Kuan-Ju Lin, Chun-Yu Liu, Wen-Liang Fang, Kuang-Yi Chang

**Affiliations:** 1Department of Anesthesiology, Taipei Veterans General Hospital, No. 201, Sec. 2, Shih-Pai Rd, Taipei 112201, Taiwan; kai7234930@gmail.com (F.-K.H.); wkchang@vghtpe.gov.tw (W.-K.C.); atmoonsamo@gmail.com (K.-J.L.); 2School of Medicine, National Yang Ming Chiao Tung University, Taipei 112304, Taiwan; cyliu3@vghtpe.gov.tw (C.-Y.L.); wlfang@vghtpe.gov.tw (W.-L.F.); 3Division of Transfusion Medicine, Department of Medicine, Taipei Veterans General Hospital, Taipei 112201, Taiwan; 4Department of Surgery, Taipei Veterans General Hospital, Taipei 112201, Taiwan

**Keywords:** blood transfusion, disease-free survival, dose-response, survival analysis, stomach neoplasms

## Abstract

**Simple Summary:**

Stomach cancer is a common malignancy and one of the leading causes of cancer death in Taiwan. Although tumor characteristics are the main determinants of oncological outcomes, modulation of the immune system may also play some role in cancer progression. Despite the hypothesis that perioperative blood transfusion may suppress the immune reactivity and promote tumor recurrence, the role of perioperative transfusion in the prognosis of stomach cancer remains controversial. To fill this gap, we designed this retrospective study using sound analytical approaches to investigate the impact of perioperative transfusion on oncologic outcomes after curative resection for stomach cancer. We demonstrated that perioperative transfusion was associated with inferior disease-free and overall survival after stomach cancer surgery and a dose-response relationship was also noted between the amount of transfusion and risk of cancer recurrence or mortality.

**Abstract:**

Background: Whether perioperative packed red blood cell (pRBC) transfusion is associated with inferior long-term outcomes after stomach cancer surgery remains controversial. Methods: This research used a retrospective cohort study. Patients with stage I~III stomach cancer undergoing tumor resection were collected at a tertiary medical center. Patient characteristics, surgical features and pathologic findings were gathered from an electronic medical chart review. The associations of perioperative pRBC transfusion with postoperative disease-free and overall survivals were evaluated using Cox regression analysis with an inverse probability of treatment weighting (IPTW). Restricted cubic spline functions were employed to characterize dose-response relationships between the amount of transfusion and cancer outcomes after surgery. Results: Among the 569 patients, 160 (28.1%) received perioperative pRBC transfusion. Perioperative transfusion was associated with worse disease-free survival (IPTW adjusted HR: 1.42, 95% CI: 1.18–1.71, *p* < 0.001) and overall survival (IPTW adjusted HR: 1.27, 95% CI: 1.05–1.55, *p* = 0.014). A non-linear dose-response relationship was noted between the amount of transfusions and worse disease-free or overall survival. Conclusions: Perioperative pRBC transfusion was associated with worse disease-free and overall survival after stomach cancer surgery, and strategies aiming to minimize perioperative transfusion exposure should be further considered to reduce the potential risk.

## 1. Introduction

Stomach cancer is a common malignancy and one of the leading causes of cancer death in Taiwan [1]. Surgical resection is still the most definitive treatment for patients with stomach cancer [2]. Although the factors that influence the recurrence of stomach cancer are mainly associated with tumor characteristics, the immune system also plays critical roles in the oncological outcomes after surgery for stomach cancer [3,4]. Blood transfusion may suppress the activity of T cell and attenuate the immune reactivity, which may increase the cancer-related mortality rate [5]. Patients with stomach cancer often received perioperative pRBC transfusion due to preoperative anemia or surgical blood loss [6]. Despite the fact that some of the literature has suggested that perioperative pRBC transfusion is an independent risk factor for worse outcomes in patients undergoing curative resection for gastric cancer [7,8,9], other studies have proposed that the negative effects may result from selection bias or uncontrolled confounders since those patients who needed perioperative transfusion were often those with worse medical conditions or a more advanced cancer stage [10,11].

In order to elucidate the relationships between perioperative pRBC transfusion and oncologic outcomes after stomach cancer surgery, we conducted this retrospective cohort study and applied robust statistics to a detailed list of clinicopathologic features associated with oncologic outcomes to isolate the effects of perioperative transfusion on recurrence and survival and determine if dose influences outcomes. We used inverse probability of treatment weighting (IPTW) based on a propensity scoring model to eliminate imbalances in patient characteristics that might affect outcomes and preserve sample size for better precision. In light of the positive findings observed in previous studies, we hypothesize that perioperative pRBC transfusion increases the risk of stomach cancer recurrence and mortality after curative surgery. We also suggest that the amount of blood given could modify the relative risk further. The purpose of our analysis is to provide health care providers with detailed information about the transfusion risk to guide clinical decision making in the perioperative period.

## 2. Materials and Methods

### 2.1. Patient Selection

After the approval of Institutional Review Board of Taipei Veterans General Hospital (IRB-TPEVGH No.: 2017-01-021AC), we reviewed the electronic medical database of all patients who received surgery for stomach cancer at a medical center from January 2009 to December 2014 and followed up them until December 2020. Those with repeated surgery, palliative surgery or metastasis, resection for benign lesion, missing demographic data, surgical and anesthetic features, and pathologic records were excluded (Figure 1). The remaining patients were classified into two groups based on whether they received perioperative pRBC transfusion or not.

### 2.2. Data Collection

The specialist anesthesiologist who reviewed the electronic medical records was not involved in data analysis. We collected the following data: age, sex, height, weight, body mass index (BMI), underlying disease based on Charlson comorbidity index (myocardial infarction, congestive heart failure, peripheral vascular disease, cerebrovascular accident or transient ischemic attack, dementia, chronic obstructive pulmonary disease, connective tissue disease, peptic ulcer disease, liver disease, diabetes mellitus, hemiplegia, chronic kidney disease, solid tumor, lymphoma, leukemia, acquired immune deficiency syndrome) [12]; preoperative hemoglobin and albumin level; pretreatment carcinoembryonic antigen (CEA) and carbohydrate antigen 19-9 (CA19-9) level [13]; anesthesia time, surgical blood loss, epidural analgesia usage, previous abdominal surgery or not, operation date, surgical technique (open or laparoscopic), operation type (total or subtotal); perioperative pRBC transfusion (transfusion within 14 days before operation, during operation, 7 days after operation) [9]; diameter of tumors, tumor nodes metastasis staging [14], histological differentiation, lymphovascular invasion, residual on surgical margin or not; whether postoperative adjuvant chemotherapy was received. The date of death was determined according to medical record or death certificate. Recurrence was decided by the radiologists and general surgeons based on the imaging studies (e.g., computerized tomography, magnetic resonance imaging, bone scan and so on).

The primary outcome was disease-free survival (DFS), which was defined as the time from the date of surgery to the date of tumor recurrence. Patient death without recurrence was treated as a censored case. The secondary outcome was overall survival (OS), defined as the time from the date of surgery to the date of death. For those without cancer recurrence or death, the survival time was regarded as the censored observations.

### 2.3. Statistical Analysis

The frequency of transfusion is presented in the Appendix A. All patients were divided into two groups based on whether they received perioperative pRBC transfusion or not. Continuous and categorical variables are presented as mean with standard deviation and count with percentage, respectively. Logarithmic transformation was conducted to reduce skewness of non-normal continuous variables (anesthesia time and blood loss during surgery). The standardized difference (SDD) was used to assess balance in the distributions of observed covariates between the two groups because it compares the difference in means in units of the pooled standard deviation and is not affected by sample size [15]. SDD can also be used to evaluate balance in observed variables between the two groups when different weights are assigned to their subjects, and an absolute SDD below the threshold of 20% was considered as low imbalance [16]. For the technical details of calculating SDD, refer to the previous literature [15]. Kaplan–Meier method was used to compare the DFS and OS curves, and occurrence of a second primary cancer between groups. Univariate Cox regression analysis was employed to evaluate the covariate effects on DFS or OS, and occurrence of a second primary cancer. Restricted cubic spline functions with three knots (placed at 2, 4 and 6 units of pRBC transfusion) were employed to evaluate the potential dose-response effects of transfusion on outcomes of interest [17]. The potential confounding effects of observed variables was minimized by weighting patients with the inverse probability of receiving perioperative transfusion to create a synthetic sample with more balanced covariates between the groups. The individual probability of receiving perioperative pRBC transfusion was obtained from the logistic regression analysis of receiving perioperative pRBC transfusion on a list of collected variables Appendix A. The inverse of treatment probability was also applied to further weighted regression analysis, and one percent of cases at the end of weighting distribution were truncated to diminish the impact of large weights on analytical results. Weighted Cox regression analysis was used to examine the association between perioperative pRBC transfusion and cancer recurrence, overall survival or occurrence of a second primary cancer based on IPTW. For sensitivity analysis, significant predictors of DFS or OS in univariate analysis were considered as candidates for stepwise model selection procedures in multivariable models. The associations between perioperative pRBC transfusion and oncologic outcomes were further examined adjusting for the determined predictors of the multivariable models. In addition, we used the quintiles of propensity scores to further classify the patients into five equal size subgroups, and stratified Cox regression analysis was conducted to obtain a pooled hazard ratio to evaluate the relationship between perioperative pRBC transfusion and DFS or OS. Moreover, multivariable logistic regression analysis with stepwise forward model selection processes was performed to determine the independent predictors for perioperative transfusion. The significance level for all hypotheses was 0.05 for a two-tailed test. All the statistical analyses were performed using SAS software, version 9.4 (SAS Institute Inc., Cary, NC, USA).

## 3. Results

Among the 569 patients included in the analysis, 160 (28.1%) received perioperative pRBC transfusion for stomach cancer resection. The median follow-up interval for all patients was 59.8 months (interquartile range: 22.8~90.7). Note that in the original dataset, more patients in the perioperative pRBC transfusion group tended to have lower preoperative hemoglobin and albumin levels, less laparoscopic surgery, longer anesthesia time and more advanced cancer (Table 1). Appendix A illustrates the absolute SDD for each of the 21 collected variables. Note that the absolute SDD before IPTW exceeded 20% for 13 (62%) of the 21 collected variables, but these imbalances were greatly improved after IPTW. For example, the largest absolute SDD before IPTW was 131.1% (preoperative hemoglobin), and its value reduced to 9.6% after weighting.

### 3.1. Perioperative pRBC Transfusion and Disease-Free Survival

In the univariate analysis, perioperative pRBC transfusion was associated with inferior DFS (crude hazard ratio (HR) = 2.86, *p* < 0.001, Figure 2a). After IPTW, the association between perioperative pRBC transfusion and postoperative cancer recurrence remained significant (adjusted HR = 1.42, 95% confidence interval (CI) 1.18 to 1.71; *p* < 0.001) in the weighted Cox regression analysis. With respect to the dose-response relationship, the IPTW-adjusted method using restricted cubic spline transformation revealed a significantly non-linear association between perioperative pRBC transfusion and cancer recurrence after surgery (Table 2). Note that the risk of recurrence increased gradually along with the increment of pRBC transfusion, peaked around 4 U and then declined slowly (Figure 3a). For sensitivity, multivariable analysis identified eleven independent predictors of DFS after stomach cancer recurrence, including sex, body mass index (BMI), Charlson comorbidity index and so on (Table 3). Note that the association of perioperative pRBC transfusion with DFS after surgery for stomach cancer was significant (adjusted HR = 1.66, 95% CI: 1.19~2.3, *p* = 0.003) as well as after the adjustment for the other significant predictors. The propensity score-based quintile method also identified a significant association between perioperative pRBC transfusion and DFS after stomach cancer surgery (pooled HR = 1.74, 95% CI: 1.2 to 2.5, *p* < 0.001)

### 3.2. Perioperative pRBC Transfusion and Overall Survival

In the univariate analysis, perioperative pRBC transfusion was associated with worse OS after surgery for stomach cancer (crude HR = 2.79, *p* < 0.001, Figure 2b). Moreover, weighted Cox regression analysis demonstrated a significant relationship between perioperative pRBC transfusion and OS (adjusted HR: 1.27, 95% CI: 1.05–1.55, *p* = 0.014) after IPTW. A significantly non-linear dose-response relationship was identified between the amount of transfusions and OS, and the peak effect was noted around 4 U (Table 2 and Figure 3b). For sensitivity analysis, multivariable regression analysis identified nine independent factors of OS, including BMI, Charlson comorbidity index, carcinoembryonic antigen and so forth (Table 4). The effect of perioperative pRBC transfusion on OS after stomach cancer surgery was significant after the adjustment for the other significant predictors (adjusted HR = 1.95, 95% CI: 1.48 to 2.58, *p* < 0.001). A significant association between perioperative pRBC transfusion and OS after stomach cancer surgery was also demonstrated using the quintile method based on the propensity score (pooled HR = 1.67, 95% CI: 1.14 to 2.44, *p* < 0.001).

### 3.3. Predictors for Perioperative Transfusion

Four independent predictors of perioperative pRBC transfusion were identified from the multivariable logistic regression analysis, including hemoglobin level, surgical blood loss, Charlson comorbidity index and BMI (Table 5).

## 4. Discussion

Our study revealed that perioperative pRBC transfusion was an independent predictor of disease-free and overall survivals in patients receiving stomach cancer surgery, and a non-linear dose-response relationship was noted in contrast to some previous studies that proposed that more transfusion would lead to more cancer recurrence and mortality [18]. In this study, we collected the major prognostic factors of cancer outcomes and applied sound statistical analysis to eliminate the potential confounding effects and provide insight into the association between perioperative transfusion and oncological outcomes after stomach cancer surgery [3,19,20,21]. The Charlson comorbidity index was included in the analysis since comorbidity is also an important predictor of cancer survival [22]. The IPTW methodology could minimize the imbalances in patient characteristics between the groups to obtain more accurate and precise estimated effects of perioperative transfusion on oncological outcomes. This approach also tends to preserve sample size and statistical power better than the traditional propensity score matching [15,23]. In addition, multivariable regression models were applied to ensure consistency of the estimated results and explore the risk factors of cancer recurrence and death after stomach cancer surgery. Moreover, restricted cubic spline functions were employed to provide a more comprehensive view of the dose-response contours between the amounts of perioperative transfusion and cancer outcomes [17]. All of these efforts aimed to generate more valid and reliable estimates and provide new evidence for a causal relationship between perioperative pRBC transfusion and inferior disease-free or overall survival after stomach cancer surgery.

The mechanism of perioperative pRBC transfusion influencing cancer recurrence may be attributed to the modulation of the immune system [24]. Perioperative pRBC transfusion may suppress interleukin-2 production, interferon gamma, natural killer cell function and monocyte activity [25]. It also promotes the immunosuppressive prostaglandins and regulatory T cell function, which enhance the inhibition of T cell activity. In addition, perioperative pRBC transfusion stimulates tumor growth, the invasion of tumor cells and malignant transformation by increasing the IL-6, vascular endothelial growth factor and hepatocyte growth factor [25]. Elmi and colleagues analyzed the American College of Surgeons National Surgical Quality Improvement Program database including 2884 patients with stomach cancer and revealed that those with perioperative pRBC transfusion had higher 30-day mortality, major mortality, postoperative complications and a longer length of hospital stay [8]. Kanda et al. collected 250 patients with stage II/III stomach cancer undergoing curative resection and reported that perioperative pRBC transfusion was related to worse recurrence-free survival and higher peritoneal recurrence rate [9]. Agnes et al. collected 38 articles for meta-analysis that disclosed significant associations between perioperative pRBC transfusion and OS, DFS, disease-specific survival and postoperative complications [7]. Lange et al. revealed that transfused patients had worse DFS and OS [26]. To explore the effect of transfusion-related immunomodulation, they compared leukocyte-depleted blood with non-leukocyte-depleted blood, which disclosed no difference in oncological outcome between the two groups [26]. Therefore, the roles of these cytokines and the immune system remain unclear. In contrast, Xiao et al. collected 1020 patients with stage II/III stomach cancer who received a gastrectomy and showed that perioperative pRBC transfusion did not influence OS and DFS [11]. However, the preoperative hemoglobin and intraoperative blood loss between groups were not balanced in their propensity score matching analysis. Rausei et al. reported that transfused patients had a worse prognosis but proposed that perioperative pRBC transfusion seemed to be a confounding factor [10]. However, the relatively small sample size and differences in covariate distributions between the groups were major limitations. The dose-response relationship has also been noted in the previous literature. In a meta-analysis study, all-cause mortality was significantly higher in patients who received >800 mL blood transfusion [27]. Moreover, a retrospective study revealed the patients transfused with >5 U had a worse prognosis than those transfused with <5 U or non-transfused [28]. Although the relationships between perioperative pRBC transfusion and oncological outcomes remain controversial, prudent perioperative transfusion management is still of clinical importance among patients receiving stomach cancer surgery.

BMI is a protective factor in our study. The previous literature revealed the U-shaped association between BMI and the oncological outcomes of stomach tumor [29]. Patients with a higher BMI had thickened visceral fat, which increased the surgical difficulty and resulted in longer operation time, fewer lymph node dissections and higher postoperative complications [30]. Moreover, the pro-inflammatory effect of obesity may accelerate the progression of the inflammatory state and malignancy transformation [29]. On the contrary, worse immune-nutritional condition and deeper tumor invasion were reported in patients with lower BMI [30]. The BMIs of our study population were mainly within the underweight to normal range, which may be a reason for the protective effect of BMI in both DFS and OS.

The choice of total or subtotal gastrectomy is based on the location, size and extent of the tumor [31]. In patients with a distal stomach tumor, both total and subtotal gastrectomy can be considered and which one is preferred is still under debate [32]. Although total gastrectomy can completely remove remnant cancer, which contributes to theoretically better prognosis, both procedures provide similar survival [33]. Subtotal gastrectomy can preserve proximal stomach and cause fewer gastrointestinal symptoms and better quality of life, but the difference in quality of life seems to reduce over time [32]. On the other side, total gastrectomy has been historically considered as the surgical treatment of choice for proximal stomach cancer. Patients with proximal stomach cancer may not be diagnosed until the tumor is large enough to cause dysphagia. A later diagnosis, larger tumor and more surgical complexity lead to a poorer prognosis for proximal stomach cancer than distal cancer [34]. This may explain why those patients in our study who received a total gastrectomy had a worse prognosis than their counterparts with subtotal procedures.

There are several limitations in our study. First, potential selection bias and confounding factors cannot be totally excluded due to the retrospective design. Second, outcomes of patients lost to follow-up were difficult to obtain and the corresponding censoring time was included in the analysis instead. Third, we only evaluated the effects of pRBC transfusion on cancer outcomes but did not further investigate the influences of other blood products that may also induce a different immune response. Finally, the immune status of collected patients was not further assessed due to data unavailability.

## 5. Conclusions

Perioperative pRBC transfusion was associated with worse disease-free and overall survival in patients receiving curative surgery for stage I to III stomach cancer. In order to improve patient outcome, a systematic method should be considered to weight the relative risks of anemia compared to blood transfusion in order to guide treatment. Strategic measures to reduce surgical blood loss and perioperative transfusion will also be beneficial to long-term outcomes after stomach cancer surgery. Well-designed prospective studies are still necessary to verify the causal relationships between perioperative transfusion and oncological outcomes after stomach cancer surgery and to elucidate the underlying mechanisms in order to identify more potential interventions.

## Figures and Tables

**Figure 1 cancers-13-05438-f001:**
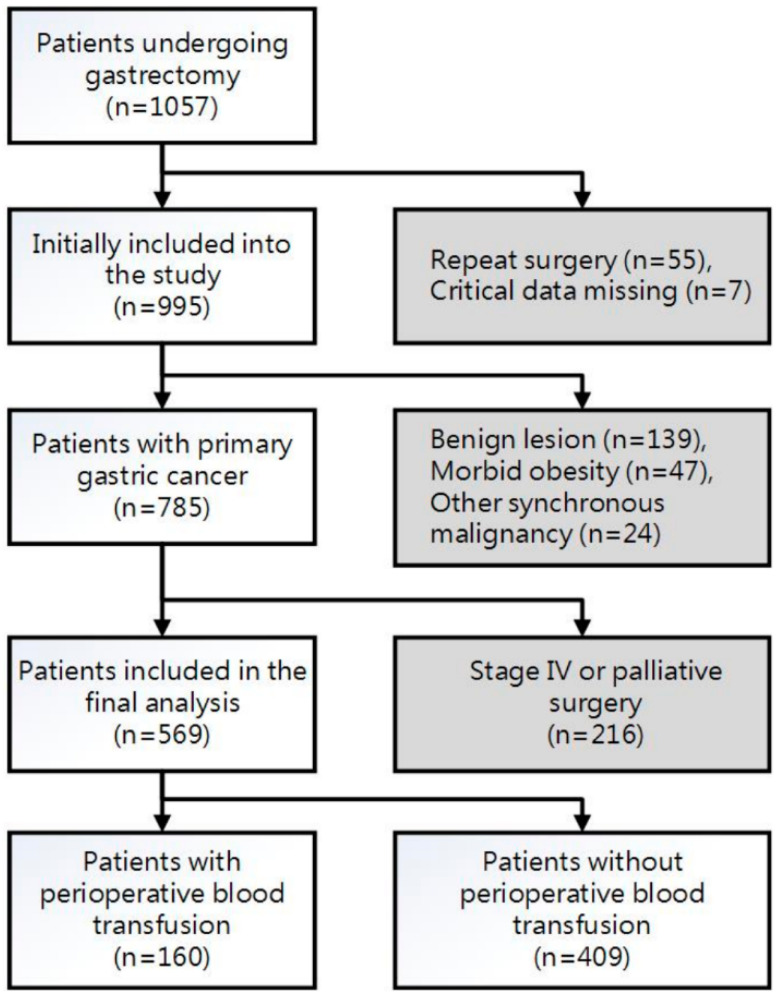
Flow diagram for patient selection.

**Figure 2 cancers-13-05438-f002:**
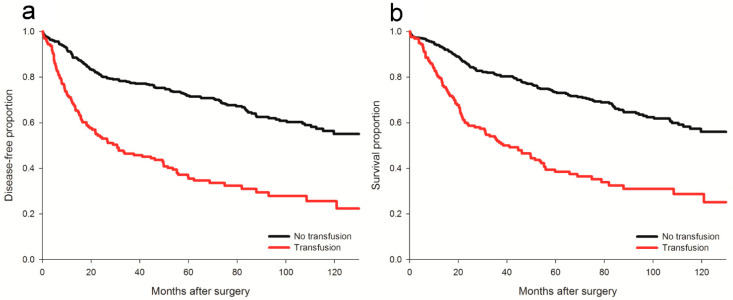
Disease-free and overall survival plots for the transfused and non-transfused patients. (**a**) disease-free survival; (**b**) overall survival. Both *p* < 0.001 by log rank test.

**Figure 3 cancers-13-05438-f003:**
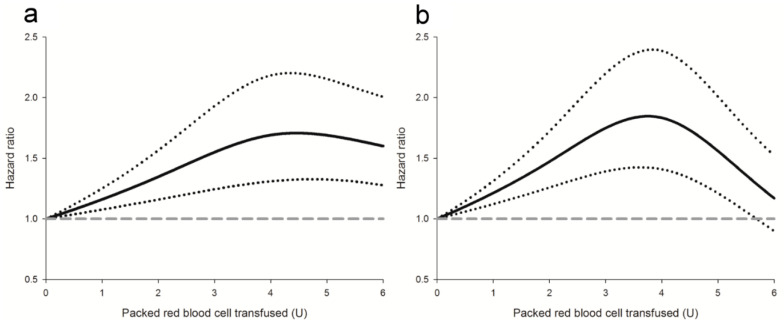
Dose-response relationships between the amount of packed red blood cell transfusions and the (**a**) disease-free or (**b**) overall survival. Solid line: hazard ratio; dotted lines: 95% confidence intervals; dashed line: a reference of hazard ratio = 1. The restricted cubic spline function used three knots located at 2, 4 and 6 units.

**Table 1 cancers-13-05438-t001:** Comparisons of patient characteristics between transfused and non-transfused patients before and after IPTW.

	Original Data			After IPTW		
Collected Variables	No Transfusion (*n* = 409)	Transfusion (*n* = 160)	SDD (%)	No Transfusion (*n* = 513)	Transfusion (*n* = 505)	SDD (%)
Sex, male	255 (62.3%)	110 (68.8%)	13.5	349 (65.3%)	384 (74.8%)	21.0
Age ≥ 70 years	181 (44.3%)	101 (63.1%)	38.5	260 (48.7%)	243 (47.3%)	2.8
Body mass index, kg·m^−2^	23.87 ± 3.57	23.53 ± 3.91	9.2	23.71 ± 3.53	23.63 ± 3.52	2.2
Charlson comorbidity index	4.7 ± 1.7	5.4 ± 1.8	43.9	4.8 ± 1.7	4.9 ± 1.8	1.5
Preoperative hemoglobin, g·dL^−1^	12.7 ±1.6	10.2 ± 2.1	131.1	12.3 ± 1.7	12.1 ± 2.5	9.6
Preoperative albumin, g·dL^−1^	4.1 ± 0.4	3.7 ± 0.5	84.7	4.0 ± 0.4	3.9 ± 0.4	18.8
CEA *	1.02 ± 1.04	1.34 ± 1.15	28.4	1.10 ± 1.11	1.15 ± 1.00	5.0
CA19-9 *	8.58 ± 0.39	8.65 ± 0.45	16.0	8.59 ± 0.39	8.65 ± 0.48	12.3
Anesthesia time, min *	7.18 ± 1.36	8.39 ± 1.43	86.4	7.43 ± 1.37	7.66 ± 1.58	15.6
Blood loss during surgery, mL *	3.06 ± 1.96	3.54 ± 2.44	21.6	3.24 ± 2.17	3.29 ± 1.88	2.3
Epidural analgesia	171 (41.8%)	70 (43.8%)	3.9	226 (42.3%)	252 (49.1%)	13.8
Previous abdominal surgery	62 (15.2%)	24 (15.0%)	0.4	86 (16.2%)	95 (18.5%)	6.1
Operation date			11.9			9.1
Before 2012	237 (57.9%)	102 (63.8%)		303 (56.8%)	268 (52.2%)	
After 2012	172 (42.1%)	58 (36.3%)		231 (43.2%)	245 (47.8%)	
Operation type			31.3			10.3
Total	82 (20.0%)	54 (33.8%)		129 (24.2%)	147 (28.7%)	
Subtotal	327 (80.0%)	106 (66.3%)		405 (75.8%)	366 (71.3%)	
Laparoscopic surgery	120 (29.3%)	15 (9.4%)	52.2	130 (24.3%)	92 (17.9%)	15.7
Stage			28.9			4.0
I	184 (45.0%)	55 (34.4%)		219 (41.0%)	220 (42.9%)	
II	113 (27.6%)	35 (21.9%)		141 (26.5%)	137 (26.8%)	
III	112 (27.4%)	70 (43.8%)		174 (32.6%)	156 (30.3%)	
Tumor size ≥ 5 cm	125 (30.6%)	79 (49.4%)	39.1	182 (34.2%)	169 (32.9%)	2.7
Histologic differentiation			21.3			24.0
Well to moderate	179 (43.8%)	87 (54.4%)		241 (45.1%)	292 (57.0%)	
Poor	230 (56.2%)	73 (45.6%)		293 (54.9%)	221 (43.0%)	
Lymphovascular invasion	171 (41.8%)	93 (58.1%)	33.1	248 (46.4%)	205 (39.9%)	13.2
Residual tumor on surgical margin	9 (2.2%)	2 (1.3%)	7.3	9 (1.8%)	3 (0.5%)	12.2
Adjunct chemotherapy	144 (35.2%)	69 (43.1%)	16.3	204 (38.3%)	215 (41.9%)	7.3

Values are presented in mean ± SD or counts (percent). Standardized difference (SDD) is the difference in mean, proportion or rank divided by the pooled standard deviation, expressed as percentage; imbalance is defined as absolute value greater than 20 (small effect size). IPTW: inverse probability treatment weighting; CEA: carcinoembryonic antigen; CA19-9: carbohydrate antigen 19-9. * On base-2 logarithmic scale.

**Table 2 cancers-13-05438-t002:** Dose-response relationship between perioperative transfusion and recurrence-free and overall survival.

	Linear Effect			Non-Linear Effect		
Outcomes	Estimate	SE	*p*	Estimate	SE	*p*
Recurrence-free survival	0.149	0.038	<0.001	−0.035	0.013	0.008
Overall survival	0.193	0.040	<0.001	−0.084	0.016	<0.001

**Table 3 cancers-13-05438-t003:** Forward model selection for disease-free survival before weighting.

Selected Variables	HR	95% CI	*p*
Packed red blood cell transfusion	1.66	1.19~2.30	0.003
Sex (F vs. M)	0.66	0.49~0.89	0.007
Body mass index	0.95	0.91~0.99	0.007
Carlson comorbidity index	1.17	1.08~1.26	<0.001
Preoperative hemoglobin	0.92	0.85~0.98	0.017
CEA *	1.14	1.03~1.27	0.012
Operation date (after 2012 vs. before 2012)	0.67	0.50~0.89	0.006
Surgery (subtotal vs. total)	0.61	0.46~0.79	<0.001
Stage			<0.001
II vs. I	1.23	0.85~1.78	0.263
III vs. I	2.21	1.60~3.06	<0.001
Tumor size > 5 cm	1.46	1.11~1.93	0.007
Residual tumor on surgical margin	3.92	1.97~7.83	<0.001

HR: hazard ratio; CI: confidence interval; CEA: carcinoembryonic antigen. * On base-2 logarithmic scale.

**Table 4 cancers-13-05438-t004:** Forward model selection for overall survival before weighting.

Selected Variables	HR	95% CI	*p*
Packed red blood cell transfusion	1.95	1.48~2.58	<0.001
Body mass index	0.94	0.90~0.97	0.001
Carlson comorbidity index	1.25	1.16~1.35	<0.001
CEA *	1.16	1.04~1.29	0.009
Operation date (after 2012 vs. before 2012)	0.67	0.50~0.90	0.008
Surgery (subtotal vs. total)	0.64	0.48~0.84	0.001
Stage			<0.001
II vs. I	1.20	0.83~1.73	0.339
III vs. I	1.98	1.43~2.76	<0.001
Tumor size > 5 cm	1.43	1.08~1.89	0.011
Residual tumor on surgical margin	3.31	1.60~6.85	0.001

HR: hazard ratio; CI: confidence interval; CEA: carcinoembryonic antigen. * On base-2 logarithmic scale.

**Table 5 cancers-13-05438-t005:** Independent predictors for perioperative blood transfusions.

Selected Variables	OR	95% CI	*p*
Hemoglobin	0.43	0.37~0.51	<0.001
Blood loss during surgery *	2.52	2.02~3.15	<0.001
Carlson comorbidity index	1.21	1.05~1.40	0.010
Body mass index	0.93	0.87~1.00	0.043

OR: odds ratio; CI: confidence interval. * On base-2 logarithmic scale.

## Data Availability

All data are available from the author directly.

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
