# Peer review of "The Associations between Perioperative Blood Transfusion and Long-Term Outcomes after Stomach Cancer Surgery"

_cancers, 2021, doi:10.3390/cancers13215438_

Round 1

Reviewer 1 Report

Dear Authors,
This manuscript entitled: "The associations between perioperative blood transfusion and long-term outcomes after stomach cancer surgery" is interesting. I provide some comments and suggestion to improve the quality of the manuscript.
1.    What is the reason why use “SDD” to present the “significance” in Table 1? 
2.    What is the reason why the age classified by 70 years old in Table 1?
3.    How to calculate linear effect and nonlinear effect (table 2)? Have the authors adjusted any variables? Or only analyze the transfusion group?
4.    The authors revealed the "Dose response relationship between perioperative transfusion and recurrence free and overall survival" in their manuscript; however, less discussion about it. Please add more discussion about this part.

Author Response

This manuscript entitled: "The associations between perioperative blood transfusion and long-term outcomes after stomach cancer surgery" is interesting. I provide some comments and suggestion to improve the quality of the manuscript.
1. What is the reason why use “SDD” to present the “significance” in Table 1? 

Ans: The standardized difference is not influenced by sample size, therefore, the use of the standardized difference can be used to compare balance in measured variables between treated and control subjects in the same sample when different weights are assigned to the same subjects. Furthermore, it allows for the comparison of the relative balance of variables measured in different units (e.g., age in years vs. systolic blood pressure in mm Hg) by calculating each on the standard deviation scale (Ref A). SDD is commonly used in propensity score-adjusted between-group comparisons in many studies. (Ref B)

Ref A: Austin & Stuart. Moving towards best practice when using inverse probability of treatment weighting (IPTW) using the propensity score to estimate causal treatment effects in observational studies. Stat Med 2015; 34(28): 3661-79. doi: 10.1002/sim.6607.

Ref B: Czarnecki et al. Association of prior β-blocker use and the outcomes of patients with out-of-hospital cardiac arrest. Am Heart J 2015; 170(5): 1018-24.e2. doi: 10.1016/j.ahj.2015.06.027.

  1. What is the reason why the age classified by 70 years old in Table 1?

Ans: Because the median of age was 70 years old, we use the median to dichotomize the distribution of age to ensure the best statistical power.

  1. How to calculate linear effect and nonlinear effect (table 2)? Have the authors adjusted any variables? Or only analyze the transfusion group?

Ans: We used restricted cubic spline functions with the IPTW-adjusted data to estimate the linear and non-linear effect after weighted Cox regression analysis. All the potential effects of collected variables have been taken into account after the IPTW and weighted Cox regression analysis. Both the estimated linear and non-linear (cubic) effects were statistically significant (Table 2) and thus non-linear combined effects on cancer recurrence and overall survival were noted as Figure 3.

  1. The authors revealed the "Dose response relationship between perioperative transfusion and recurrence free and overall survival" in their manuscript; however, less discussion about it. Please add more discussion about this part.

Ans: We have added a paragraph to address the dose-response relationships in the section of Discussion as follows “The dose-response relationship was also noted in previous literatures. In a meta-analysis study, all-cause mortality was significantly higher in patients received >800 mL blood transfusion [26]. Besides, a retrospective study also revealed the patients transfused >5 U had worse prognosis than those transfused <5 U or non-transfused [27]”. (Page 10, Line 292 to 295)

Reviewer 2 Report

Fu-Kai Hsu et al. examined the relevance of blood transfusion as a prognostic factor for overall (OS) and disease-free survival (DFS) of patients with gastric cancer. They carried out a retrospective cohort study showing that perioperative transfusion was associated with inferior disease-free and overall survival. Generally, the paper is well planned and addresses a clinically relevant topic. However, its novelty is rather limited. Three meta-analyses (Agnes et al. Eur J Surg Oncol. 2018, Li et al. Medicine (Baltimore). 2015, Sun et al. Int J Surg. 2015) and numerous later studies recruiting larger populations evaluated the same topic. Moreover, the results of these studies were similar to the findings of the present manuscript.

Minor points:

  • The disease-free survival (DFS) defined as the time from the date of surgery to the date of tumor recurrence or patient mortality is unclear. Patient’s death without recurrence should be treated as a censored case.
  • Timing of transfusion (preoperative vs postoperative) was not specified.

Author Response

Fu-Kai Hsu et al. examined the relevance of blood transfusion as a prognostic factor for overall (OS) and disease-free survival (DFS) of patients with gastric cancer. They carried out a retrospective cohort study showing that perioperative transfusion was associated with inferior disease-free and overall survival. Generally, the paper is well planned and addresses a clinically relevant topic. However, its novelty is rather limited. Three meta-analyses (Agnes et al. Eur J Surg Oncol. 2018, Li et al. Medicine (Baltimore). 2015, Sun et al. Int J Surg. 2015) and numerous later studies recruiting larger populations evaluated the same topic. Moreover, the results of these studies were similar to the findings of the present manuscript.

Ans: Although there have been some studies investigating the association of perioperative transfusion with gastric cancer outcomes, our study provides new evidence and information about their causal relationship with following strengths. First, restricted cubic spline functions were employed to provide a more comprehensive view of the dose-response contours [15] and we demonstrated “non-linear” dose-response relationships which have not been mentioned in the previous studies. Second, we took more prognostic factors of cancer outcomes into consideration and applied sound statistical analysis to eliminate the potential confounding effects and selection bias. Third, Charlson comorbidity index was included in the analysis to reduce confounding effects from comorbidities on cancer survival [20]. Fourth, the IPTW methodology was used to reduce the imbalances in patient characteristics between the groups to obtain more accurate and precise estimated effects of perioperative transfusion on oncological outcomes. This approach also tends to preserve sample size and statistical power better than the traditional propensity score matching [21,22]. Fifth, multivariable regression models were also applied to ensure consistency of the estimated results and explore the risk factors of cancer recurrence and death after stomach cancer surgery. Our analytical approaches are novel and can generate more valid and reliable analytical results. We have mentioned these in the first paragraph of Discussion (Page 10, line 243 to 263) and believe our study is of clinical and academic importance and will be of interest to the broad readers of Cancers.

Minor points:

  1. The disease-free survival (DFS) defined as the time from the date of surgery to the date of tumor recurrence or patient mortality is unclear. Patient’s death without recurrence should be treated as a censored case.

Ans: We have added this description in the manuscript (Page 3, line 107).

  1. Timing of transfusion (preoperative vs postoperative) was not specified.

Ans: The perioperative transfusion was defined as “transfusion within 14 days before operation, during operation, 7 days after operation”. We have added this sentence in the section of Materials and Methods (Page 3, line 99). Since we evaluated the associations between perioperative transfusion and long-term cancer outcomes, The specification of transfusion timing was not necessary.

Ref: Kanda et al. Adverse prognostic impact of perioperative allogeneic transfusion on patients with stage II/III gastric cancer. Gastric Cancer. 2016; 19(1): 255-63. doi: 10.1007/s10120-014-0456-x.

Round 2

Reviewer 1 Report

No more questions for the authors.

Author Response

Thank you for your thoughtful comments and suggestions.